# Severe and widespread coral reef damage during the 2014-2017 Global Coral Bleaching Event

Ocean warming is increasing the frequency, extent, and severity of tropical-coral bleaching and mortality. During 2014–2017, marine heatwaves caused the Third Global Coral Bleaching Event. We analyze data from 15,066 reef surveys globally during 2014–2017. Across all surveyed reefs, 80% and 35% experienced moderate or greater (affecting >10% of corals) bleaching and mortality, respectively. We assess the global extent of coral bleaching and mortality by applying bleaching response curves calibrated from surveyed reefs to predict bleaching globally, based on comprehensive remote-sensing of heat stress. These models predict that 51% and 15% of the world's coral reefs suffered moderate or greater bleaching and mortality, respectively, during one or multiple years, surpassing damage from any prior global coral bleaching event. Our findings demonstrate that the impacts of ocean warming on coral reefs are accelerating, with the near certainty that ongoing warming will cause large-scale, possibly irreversible, degradation of these essential ecosystems. With heat stress levels during this event surpassing those observed previously, the National Oceanic and Atmospheric Administration developed more extreme Bleaching Alert levels that are now being used during the ongoing Fourth Global Coral Bleaching Event.

Before the 1980s, mass coral bleaching and mortality events due to heat stress were rare[1]. In the last four decades, these events have become increasingly frequent and severe. Ocean warming is now the foremost threat to coral reefs worldwide[2–7], and recurrent strong marine heatwaves are causing mass bleaching of corals on regional and global scales[2,3,8,9]. Coral bleaching occurs when the relationship between corals and their photosynthetic symbionts breaks down[10]. Bleached corals are physiologically damaged, nutritionally compromised, and may die if the bleaching is severe or prolonged. From June 2014 to May 2017 (hereafter referred to as 2014–17), reefs around the world experienced the third global-scale coral bleaching event[11–13]. This event was, at that time, the most severe global heat stress event recorded on coral reef ecosystems[2,8,11,14], surpassing the two prior global coral bleaching events recorded in 1998[15] and 2010[3,16]. Moreover, 2014–17 was the first record of a global coral bleaching event lasting much beyond a single year[2,8,11,14]. Specifically, the event spanned 3 years, with bleaching at some locations continuing after the global event concluded[17–19]. Numerous studies have revealed how this event has impacted coral reefs locally at sites around the globe, including the most severe impacts on record in many locations (see[20] and references therein).

Here, we analyze the heat stress affecting coral reefs during 2014–17 and the resultant bleaching and mortality to derive statistical relationships between remotely-sensed heat stress and onsite surveys of coral bleaching and mortality at a global scale. We then use these relationships to estimate the global impact of this mass coral bleaching event, accounting for the actual distribution of heat stress across the world's reef areas. We find substantial variation in the temperature sensitivity of bleaching and survival among years and basins. Overall, we estimate that over half the world's reefs experienced moderate or greater bleaching, and 15% experienced moderate or greater mortality. The estimated levels of accumulated heat stress, bleaching, and

✉e-mail: corals.and.climate@gmail.com; scott.heron@jcu.edu.au; ConnollyS@si.edu

mortality during this event exceeded the severity from all prior global coral bleaching events.

## Results and discussion
### Three years of heat stress on reefs
Satellite remote sensing was used to identify cumulative heat stress on coral reefs, using the Degree Heating Week (DHW) product from the National Oceanic and Atmospheric Administration's Coral Reef Watch (CRW). The onset of coral bleaching and mortality was previously associated with exposure thresholds of DHW ≥ 4 °C-weeks (Alert Level 1) and ≥8 °C-weeks (Alert Level 2), respectively[5,8]. As a result of record heat stress seen during the Third Global Coral Bleaching Event, CRW established new Alert Levels 3–5 that correspond to the risk of increasingly severe bleaching and mortality of corals across reefs[21] (Supplementary Table 1). This new set of Alert Levels is now being used during the ongoing Fourth Global Coral Bleaching Event[22].

During the Third Global Coral Bleaching Event in 2014–17, 65.8% of ~5 × 5 km² satellite remote sensing pixels containing coral reefs ($n = 53,997$) experienced heat stress classified as sufficient to cause moderate or greater bleaching (affecting >10% of corals) (DHW ≥ 4 °C-weeks, Alert Level 1 or higher) (Figs. 1a, 2, Supplementary Fig. 1a, Supplementary Table 2[23,24]). This compares with 37.2% in the Second Global Coral Bleaching Event (Fig. 1b) and 20.9% in the First Global Coral Bleaching Event (Fig. 1c). Among all reef-containing pixels, 23.6% also reached or exceeded the higher threshold of DHW ≥ 8 °C-weeks (corresponding with Alert Level 2 or higher), classified as sufficient to cause severe bleaching (affecting >50% of corals) and moderate or greater mortality (affecting >10% of corals), during the Third Global Coral Bleaching Event, compared with 9.5 and 2.5% in the Second and First Global Coral Bleaching Events, respectively. Some 5.0% of reef-containing pixels reached or exceeded the new Alert Level 3 (DHW ≥ 12 °C-weeks), compared with 2.1% in the Second Global Coral Bleaching Event and 0.5% in the First Global Coral Bleaching Event, indicating

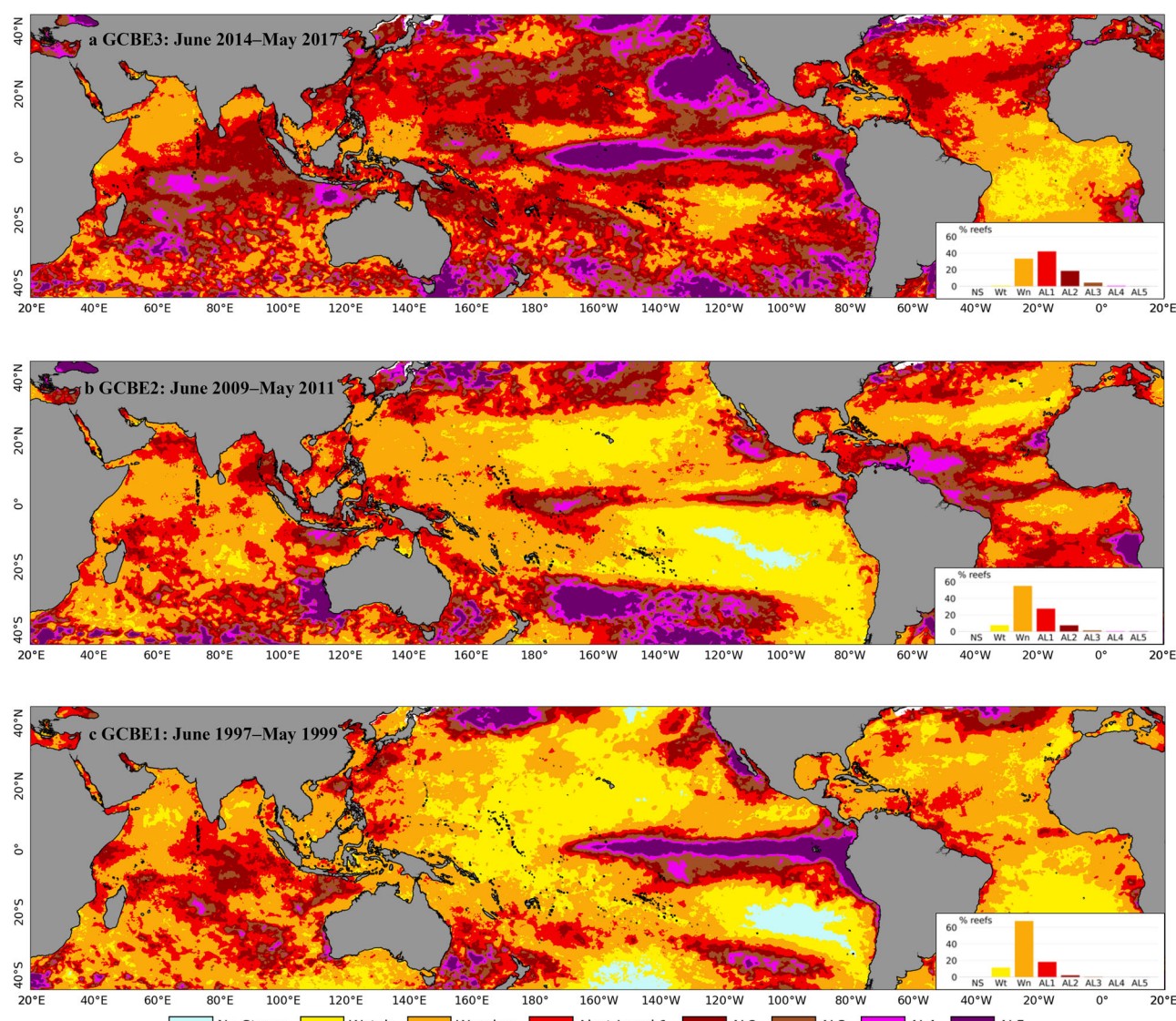

**Fig. 1 | Global distribution of heat stress from the first three Global Coral Bleaching Events.** Global pattern of maximum heat stress from **a** The Third Global Coral Bleaching Event (GCBE3) 2014–17, **b** GCBE2 2009–2011, and **c** GCBE1 1997–1999. Heat stress categories[24] of Alert Levels 1 and 2 correspond to moderate reef-wide coral bleaching (4 ≤ DHW < 8 °C-weeks) and severe reef-wide bleaching with moderate mortality (8 ≤ DHW < 12 °C-weeks), respectively. Newly established Alert Levels 3–5 correspond to the risk of increasingly severe mortality of corals across reefs (Supplementary Table 1)[21]. Inset histograms show the percentage of reef-containing pixels reaching each heat stress level during each global coral bleaching event. See Supplementary Fig. 1a for global coral reef locations. The data and code to generate this figure are provided in Zenodo (https://doi.org/10.5281/zenodo.15114357).

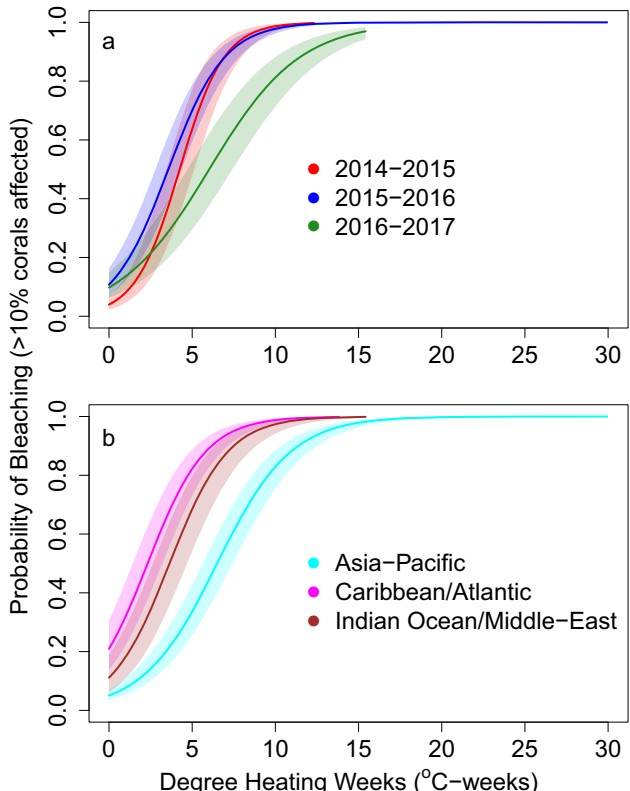

**Fig. 2 | Fitted response curves for moderate or greater coral bleaching (affecting >10% of corals) as a function of heat stress varied among years and basins.** Bleaching-response curves, with 95% confidence limits (shading), in each of the: **a** three bleaching years (2014–15, 2015–16, 2016–17) and **b** three basins Asia-Pacific (AP), Caribbean/Atlantic (CA), and Indian Ocean/Middle East (IM). The vertical axis is the probability of moderate or greater bleaching estimated from the bleaching database. Curves extend across the domain of DHW values apparent for that basin or year. Solid lines represent the mean probability of bleaching, and shaded regions represent 95% confidence intervals on the mean. The data and code to generate this figure are provided in Zenodo (https://doi.org/10.5281/zenodo.15114357).

Third Global Coral Bleaching Event, bleaching was seen in additional areas such as Hawai'i[26,27], Micronesia[28,29], the Marianas[17], and the Great Barrier Reef[30].

## Bleaching and mortality

In response to the widespread and severe heat stress, we assembled coral bleaching and mortality data from 15,066 in-water and aerial surveys from teams around the globe—inclusive of observations from a pre-existing database covering the 1960s through 2010[31,32] (Supplementary Data 1, Supplementary Fig. 1b). During the Third Global Coral Bleaching Event, 80% of surveys reported a moderate or greater level of bleaching (affecting >10% of corals—either as a percentage of the total coral cover or a percentage of colonies, see Methods), while 31% of all surveys reported that the bleaching was severe (affecting >50% of corals). Additionally, 35% of surveys reported moderate or greater (>10%) recent coral mortality, with 6% of surveys reporting severe mortality (>50%).

Our global analyses showed a strong positive relationship of bleaching and mortality with satellite-derived, accumulated heat stress (DHW), while the rate of biological response to heat stress varied among ocean basins and bleaching years (June-to-May periods). Comparing observations of moderate or greater reef-level bleaching (Fig. 2, Supplementary Fig. 3) across all basins, bleaching sensitivity to heat stress was generally highest in 2015–16 and lowest in 2016–17 (Fig. 2a), suggesting acclimatization or adaptation, due to selective mortality of vulnerable host genotypes and changes in the genetic makeup of their endosymbiotic communities[33], loss of heat-sensitive phenotypes[34] or compositional changes due to selective mortality of particular genotypes or taxa[35,36]. Across all years, bleaching sensitivity was highest in the Caribbean/Atlantic basin and lowest in the Asia-Pacific (Fig. 2b).

The models that best fit the bleaching data included interactions between DHW and both regions and years (Supplementary Tables 4, 5). The best-fit model for moderate or greater bleaching (Supplementary Fig. 3) had a prediction accuracy of 83% and showed interactions between years and basins. Corals in all years and basins showed moderate to high probabilities of bleaching in response to heat stress at levels above the 4 °C-weeks threshold normally considered predictive of moderate bleaching[24]. The Caribbean/Atlantic showed greater bleaching sensitivity than the Asia-Pacific in 2014–15 and 2015–16 but not in 2016–17. Additionally, the Indian Ocean/Middle East also showed greater bleaching sensitivity than the Asia-Pacific in 2016–2017 but was intermediate to and not clearly statistically distinguishable from the other regions' thresholds in the first two bleaching years (Supplementary Fig. 3).

The best-fitting model for severe bleaching had a predictive accuracy of 88%; and spatiotemporal patterns of severe bleaching (Supplementary Fig. 4) were similar to those seen in moderate bleaching, including greater sensitivity in 2015–16 than 2016–17 in the Asia-Pacific (Supplementary Fig. 4a), and reduced sensitivity in 2015–2016 and in 2016–17 than 2014–2015 in the Caribbean/Atlantic (Supplementary Fig. 4b). However, estimated thresholds for severe bleaching (Supplementary Fig. 4) had greater uncertainty than those for moderate bleaching (Supplementary Fig. 3), likely due to the much smaller number of observations of severe bleaching, and an uneven spatial distribution of severe bleaching. Although observations from the Caribbean were, on average, a few meters deeper than those from the other two basins (Supplementary Fig. 6), sensitivity analyses incorporating depth as a covariate showed that the among-basin and inter-annual variations in bleaching thresholds were robust to differences in depth; though, for severe bleaching, broader confidence intervals reduced the statistical evidence for some of the differences observed in the original analysis (Supplementary Figs 7 and 8).

Records of moderate or greater (>10%) coral mortality again varied among basins and years, with the risk of mortality clearly increasing

a risk of multi-species mortality. In all three global coral bleaching events, less than 1% of reef-containing pixels reached or exceeded the new Alert Level 4 (DHW ≥ 16 °C-weeks), indicating a risk of severe, multi-species mortality. Few pixels (0.1%) reached the new Alert Level 5 (DHW ≥ 20 °C-weeks), indicating a risk of near complete mortality.

Repeated heat-stress exposure was widespread during the Third Global Coral Bleaching Event. Half (47.2%) of reef-containing pixels that reached Alert Level 1 and one-fifth (20.8%) of reef-containing pixels that reached Alert Level 2 did so at least twice during 2014–17. Heat stress on reefs during the Third Global Coral Bleaching Event (Fig. 1a) was higher than that seen in either the Second (Fig. 1b) or the First Global Coral Bleaching Events (Fig. 1c).

Global coral bleaching events have followed a consistent spatial progression that was observed anecdotally during all three global coral bleaching events[10,11] and is now discernable analytically[14]. Driven by ocean heating due to El Niño and its teleconnected impacts[25], the highest sea surface temperature anomalies are typically first seen in the equatorial eastern Pacific Ocean, then anomalies appear sequentially across the equatorial central Pacific Ocean, in the western South Pacific Ocean, southern Indian Ocean, northern Indian Ocean, Middle Eastern seas and Southeast Asia, then the Caribbean Sea and western Atlantic Ocean (Supplementary Fig. 1c–f, Supplementary Video 1). During some past global coral bleaching events, as well as during the

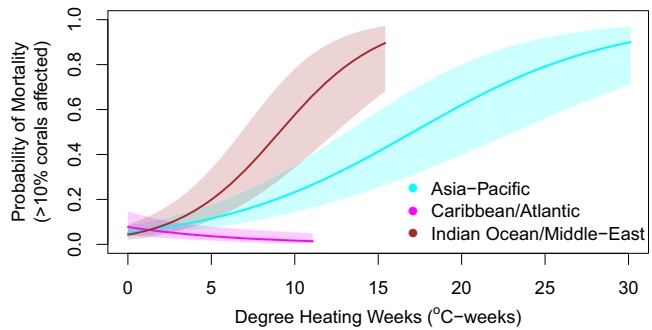

**Fig. 3 | The response curves for moderate or greater coral mortality (affecting >10% of corals) as a function of heat stress varied among ocean basins across all years.** The vertical axis is the probability of moderate or greater mortality calculated from the mortality data. Solid lines represent the mean probability of mortality, and shaded regions represent 95% confidence intervals on the mean. The data and code to generate this figure are provided in Zenodo (https://doi.org/10.5281/zenodo.15114357).

with heat stress level in the two Indo-Pacific basins (Fig. 3, Supplementary Fig. 5). Because the number of mortality reports was less than half the number of bleaching reports, mortality response curves (Fig. 3, Supplementary Fig. 5) were less well constrained than those for bleaching (Fig. 2, Supplementary Fig. 3). Nevertheless, the best-fitting model had a prediction accuracy of 88% and showed a probability of moderate mortality exceeding 0.2 at heat stress levels beyond 8 °C-weeks in both the Indian Ocean/Middle East and Asia-Pacific basins, consistent with use of this threshold as a global predictor for moderate mortality onset[5,8]. In contrast, the lack of a positive relationship between mortality and heat stress in the Caribbean/Atlantic in any year (Supplementary Fig. 5b) suggests that heat stress was probably not a primary driver of mortality in the Caribbean/Atlantic during 2014–17 (discussed below). As with bleaching, the among-basin and interannual patterns in mortality were robust to the slight differences in the distribution of observation depths across ocean basins (Supplementary Fig. 9).

In 2016, over 15% of global reef containing pixels, including large areas of the tropical Pacific, the Indo-Pacific, and some of the Northwestern Hawaiian Islands, reached or exceeded heat stress of 16 °C-weeks (Fig. 1). Surveys conducted in these areas showed most reefs suffered rapid and severe mortality of many coral species[37]. This heat stress corresponded with very high probabilities of moderate bleaching (Fig. 2, Supplementary Fig. 3), and greater than 50% probabilities of severe bleaching (Supplementary Fig. 4) and moderate mortality (Fig. 3, Supplementary Fig. 5). Establishment of new Alert Levels up to Level 5 (DHW ≥ 20 °C-weeks) will help capture the effects of increasingly long and intense marine heatwaves[8,9] that became especially apparent during the Third Global Coral Bleaching Event and are predicted to become more frequent in the future[38]. In contrast to the recent past, the tropical marine heatwave analyzed herein spanned multiple seasons and persisted for more than a year on some equatorial reefs (e.g., record heat stress in 2014–17 at Jarvis Island in the central Pacific Ocean resulted from over 12 months of heat stress exposure, with DHW ≥ 4 °C-weeks spanning March 2015–May 2016[39,40]).

**Assessing the global footprint of the 2014–17 bleaching**
The prediction accuracy obtained in our analyses was high overall (83–88%), and much of that prediction accuracy was due to the fixed effects components of our models (that is, the variation predicted by DHW, ocean basin, and bleaching year: 73% moderate or greater bleaching, 83% severe bleaching, and 81% moderate or greater mortality). These values compare favorably with earlier analyses of regional-scale mass bleaching events[7,30,41,42]. However, simply reporting

percentages of observed bleaching or mortality from surveyed reefs, as in previous global studies[6,43,44], could lead to overestimates of severity if reefs experiencing higher heat stress were disproportionately surveyed. Therefore, to correct for such a bias, we used our statistically modeled bleaching and mortality relationships for each basin and year to predict bleaching and mortality on each reef-containing pixel based on the maximum satellite-measured heat stress at that pixel. From this analysis, we estimated that 51% of global coral reef locations suffered moderate or greater bleaching and 15% suffered moderate or greater mortality. While the Third Global Coral Bleaching Event progressed globally for three full years (June 2014–May 2017), bleaching and mortality varied in time and space in relation to heat stress across the 21 GCBE3 regions in each year (Fig. 4, Supplementary Table 3). For example, the Coral Triangle (region #6) experienced relatively low-level heat stress in 2014–15 (DHW = 4.7 °C-weeks, yellow diamond) with an estimated 7% bleaching, compared with the highest heat stress (10.3 °C-weeks, dark red) and 32% bleaching in 2015–16, and an intermediate level of heat stress (9.9 °C-weeks, red) and 22% bleaching in 2016–17. In contrast, peak heat stress in the Great Barrier Reef (region #18) increased through the three bleaching years—4.4, 8.3, and 10.8 °C-weeks, respectively—whereas bleaching was highest in the second year (5, 49, and 35%, respectively), potentially revealing acclimatization and/or mortality of sensitive taxa.

In general, higher heat stress resulted in higher bleaching and mortality throughout the event, and most GCBE3 regions followed this pattern (Fig. 4). More complex patterns in the relationship between bleaching and mortality emerge in some areas. Heat stress-driven mortality in most basins was modeled to occur in 30–50% of pixels where bleaching occurred. However, corals in much of the Caribbean/Atlantic that bleached at similar heat-stress levels as other basins (Fig. 2, Supplementary Fig. 3) showed lower new mortality, with fewer than 15% of modeled "bleaching" pixels predicted to have experienced moderate mortality. This low mortality may have resulted from a long history and higher frequency of heat stress events in the Caribbean/Atlantic[2,3,8]. Caribbean/Atlantic coral losses have been quite high in recent decades[44,45], including bleaching-related mortality during both prior global bleaching events[15,16], the 1982–83 bleaching event[46], and the 2005 Caribbean bleaching event[5]. Additionally, coincident with the Third Global Coral Bleaching Event, disease-related mortality due to Stony Coral Tissue Loss Disease was extensive throughout the Florida Reef Tract but was not reported elsewhere in the Caribbean/Atlantic until after May 2017[47]. Heat stress was low in Florida relative to other parts of the basin in the Third Global Coral Bleaching Event. Thus, we suspect that the slight negative relationship seen between heat stress and mortality (Fig. 3, Supplementary Fig. 5b) was an artifact of the geographic distribution of Stony Coral Tissue Loss Disease (Florida-only during this event) overlaid upon the pattern of heat stress during the Third Global Coral Bleaching Event (low heat stress in Florida).

Together with impacts from overfishing, land-based pollution, and sea urchin diseases, bleaching- and disease-related mortality have reduced Caribbean/Atlantic coral density and cover[45], particularly among acroporid corals[48], as well as diversity[49]. Depauperate in species diversity for millennia, the decline in sensitive coral species since the 1980s left many Caribbean/Atlantic reefs with especially stress-tolerant species and genotypes that survive bleaching[49] but provide diminished ecosystem function[50,51]. One exception to this pattern was the unique coral assemblages off the coast of Brazil. Despite being the most biodiverse reef complex in the South Atlantic, these corals experienced low bleaching and mortality during the Third Global Coral Bleaching Event, likely due to reduced irradiance afforded by turbidity in Brazil's shallow coastal waters[52]. Unfortunately, these corals suffered major losses due to heat stress events after 2017[53].

The heat stress on coral reefs during the Third Global Coral Bleaching Event was much greater than that reported in either the

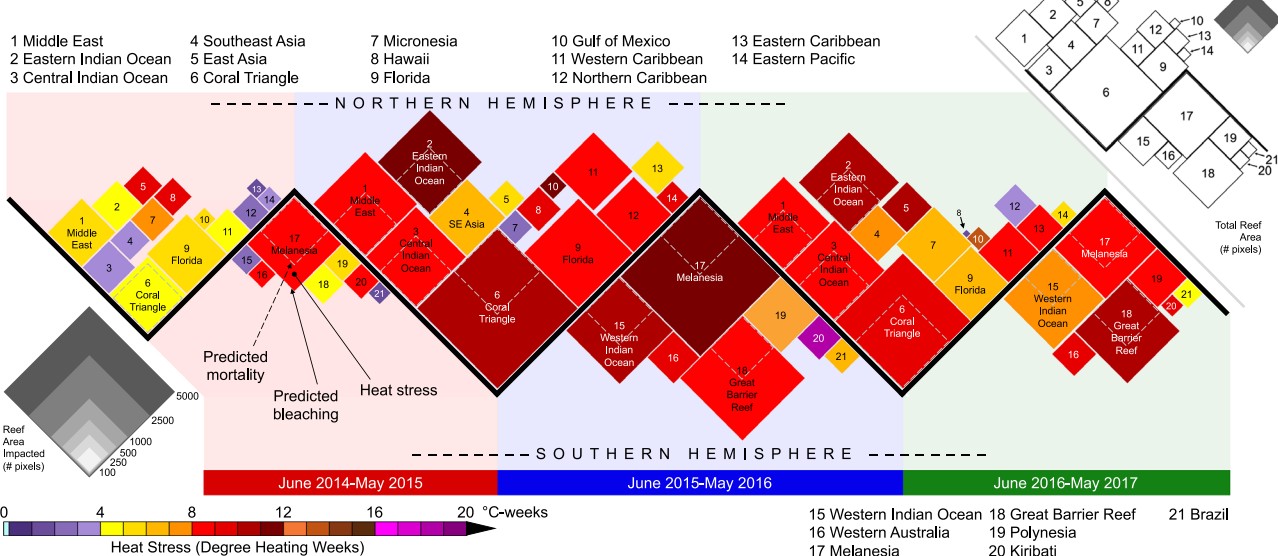

**Fig. 4 | Projected coral bleaching and mortality impacts during the Third Global Coral Bleaching Event (GCBE3).** Model-predicted extent of reefs impacted by moderate or greater (>10% of corals) bleaching (solid diamonds) and moderate or greater mortality (sub-diamonds, outlined in white inside the larger diamonds) for the 21 GCBE3 regions (Supplementary Data 1), in units of number of -5 × 5 km² satellite pixels (scale at left). The central solid-black line denotes the Equator through annual cycles. Colors represent the maximum heat stress in each region in that bleaching year (DHW in units of °C-weeks, scale at lower-left); to improve legibility, white text was used in diamonds for heat stress below 2 °C-weeks and above 9 °C-weeks; black text was used for 2−9 °C-weeks. The inset (top-right) indicates the total reef area for each GCBE3 region (scale divisions in the inset correspond with the area scale at the lower left of the main image). Predicted mortality is shown only where the area exceeded 500 pixels (-0.1% of global reef pixels). The Eastern Atlantic region was not shown due to a small reef area (4 of 53,997 reef pixels). Heat stress values for reef-containing pixels used to estimate location-specific bleaching and mortality are provided in Supplementary Table 3. The data and code to generate this figure are provided in Zenodo (https://doi.org/10.5281/zenodo.15114357).

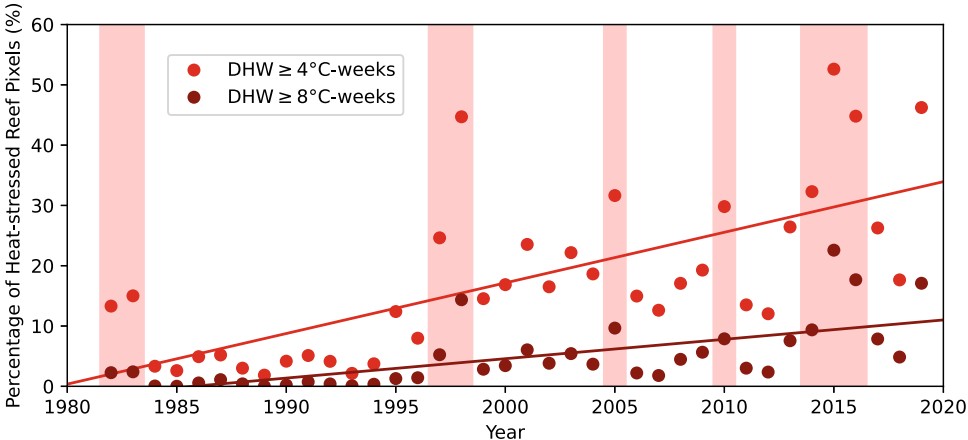

**Fig. 5 | Percentage of global reef pixels reaching DHW ≥ 4 and 8 °C-weeks calculated from the NOAA Optimum Interpolation Sea Surface Temperature (OISST), Version 2.1[67].** Years on the horizontal-axis correspond to the first year of each bleaching-year couplet (i.e., 2014 = June 2014−May 2015). Pink boxes correspond to major El Niño-Southern Oscillation events. The data and code to generate this figure are provided in Zenodo (https://doi.org/10.5281/zenodo.15114357).

1997−1999 First or 2009−2011 Second Global Coral Bleaching Events[6,31,32,44]. Using DHW calculated from a dataset covering 1982 to present, heat stress during the 2014−17 event (and even in 2016 alone) was more intense and widespread than prior mass bleaching events (Fig. 5). This resulted in greater bleaching and mortality during the Third Global Coral Bleaching Event than those earlier, less-intense events, especially given that elevated levels of bleaching and mortality were reported in successive years[6,20].

Some reefs damaged during June 2016−May 2017 continued to experience bleaching later in 2017[17,18], and others were severely bleached in subsequent years[19,30,53]. With so little time for recovery between consecutive heat stress events, many coral reefs worldwide are becoming increasingly susceptible to ecosystem collapse[35] or severe population bottlenecks (frequent intervals of low population size, with concomitant loss of genetic diversity and increased risk of local extinctions), as well as suffering knock-on effects on herbivores and other reef-associated biota[54–56]. For example, recovery of coral cover on the Great Barrier Reef was reported through 2022 following recent bleaching events[57]. While encouraging, this was driven by fast-growing *Acropora* corals that are highly susceptible to bleaching during subsequent heat stress, wave damage from storms and tropical cyclones, and crown-of-thorns starfish predation. This "recovery" measured by coral cover, may be masking a concomitant loss of coral diversity and associated functional traits[51]. Unfortunately, bleaching in

2022 ended that recovery trend[58] and in 2023–24, initial analysis reported moderate or greater (>10%) bleaching of 73% of survey reefs on the Great Barrier Reef[59], with major losses expected. With projected increases in the frequency and duration of heat stress[3,38], boom-and-bust cycles in coral cover may trigger the transition of reef ecosystems from marine biodiversity hotspots to new, lower diversity stable states[60] characterized by only a few thermally tolerant species; similar to the current state of most Caribbean/Atlantic reefs.

The 2014–17 Third Global Coral Bleaching Event was more widespread and damaging than any prior bleaching event on record, highlighting the threat of increasingly severe and widespread marine heatwaves to coral reefs that is outpacing the capacity of corals to physiologically resist heat stress[6,30,61]. The extent, frequency, and severity of marine heatwaves on coral reefs[2,31] and elsewhere[9,62] are predicted to intensify further[63,64]. One consequence of increasingly frequent bleaching events is a reduction in the time available between events for coral and ecosystem recovery[6], leading to a decline in the structure of reefs. As coral reefs are ecosystem builders that protect shorelines from wave-driven flooding and erosion, and provide food, medicines, cultural identity, and livelihoods for over a billion people[65], these essential ecosystems urgently need protection. Our study found severe coral reef losses throughout this protracted three-year bleaching event, an acceleration of the threat that has already placed them as one of the world's most sensitive ecosystems to temperature extremes[66]. While the Third Global Coral Bleaching Event was unprecedented at the time, the observed impacts may be equaled or surpassed in subsequent events, including the ongoing Fourth Global Coral Bleaching Event[22]. As ocean warming continues, coral loss and associated reef degradation around the world are nearly certain to accelerate.

## Methods
### Satellite-based heat stress
To assess heat stress that can lead to coral bleaching and mortality, we used daily global 0.05° (-5 × 5 km$^2$) sea surface temperature (SST) and heat stress metrics from the U.S. National Oceanic and Atmospheric Administration's CRW version 3.1 Satellite Coral Bleaching Heat Stress Monitoring Product Suite dataset[23,24]. The primary metric of heat stress, DHW, accumulates instantaneous bleaching heat stress during the most recent 12-week period and has been associated with risk of bleaching (Alert Level 1) and mortality (Alert Level 2)[24] (Supplementary Table 1). As a result of record heat stress seen during the Third Global Coral Bleaching Event and in subsequent years, CRW established new Alert Levels 3–5, corresponding to progressively higher threshold DHW values[21] (Supplementary Table 1 and main text). Further explanation and justification for each Alert Level is located at: https://coralreefwatch.noaa.gov/product/5km/index_5km_baa-max-7d.php.

With the Third Global Coral Bleaching Event beginning in June 2014, bleaching years, as defined for this paper, covered June through May for 2014–15, 2015–16, and 2016–17. The June–May timing also incorporates the mid-year lull in heat stress and bleaching observations that is apparent at almost all reef locations[3]. Maximum Bleaching Alert Level values for the entire period of the Third Global Coral Bleaching Event were plotted in Fig. 1 and Supplementary Fig. 1a (the latter containing an overlay of reef-containing satellite pixels), and a summary of the number of pixels reaching Alert Levels 1–5 by bleaching year was summarized (Supplementary Table 2). Maximum Bleaching Alert Level values for the First and Second Global Coral Bleaching Events were plotted in Fig. 1b,c. Because the First and Second Global Coral Bleaching Events did not follow the same June-May heat stress timing as the Third Global Coral Bleaching Event[14], 2-year couplets of June 1997–May 1999 and June 2009–May 2011 were plotted to capture all of the peak heat stress seen during these events. Corresponding charts of maximum DHW for the global coral bleaching events were plotted in Supplementary Fig. 2. Remote sensing-derived data were extracted, calculated, and plotted using IDL 8.7 and Python 3.8.5, with the following required Python computing environment: netcdf4 version 1.5.5.1, scipy version 1.6.0, python version 3.8.5, matplotlib version 3.3.3, pytables version 3.6.1, basemap version 1.2.2, basemap-data-hires version 1.2.2, pandas version 1.2.3, xarray version 0.20.1, pooch version 1.4.0, scikit-learn version 1.0.2, numba version 0.55.0, pyts version 0.12.0.

We extracted the daily and bleaching year maximum DHW values for the pixel co-located with each field survey. Surveys undertaken prior to the maximum heat stress were compared with the contemporary DHW value, while those undertaken on or after the date of the maximum heat stress were compared with the maximum value. For observations that contained an observation month without a date, we assigned a value of the 15th of the month. For onsite observations on or next to coastlines, we extracted DHW values from the nearest valid ocean pixels. For Fig. 5, global Bleaching Alert Levels were calculated from the 0.25° (-25 × 25 km$^2$) Optimum Interpolation Sea Surface Temperature (OISST), Version 2.1 dataset[67,68] using the same methods as CRW's operational products[24] and extracted for June–May bleaching years and plotted with Microsoft Excel. OISST was chosen for this global application as, at the time this article was written, the higher resolution CRW dataset[23] did not yet extend back to 1982.

### Bleaching and mortality observations
We collated field observational surveys conducted from 2014 through 2017 to document the spatial extent and severity of bleaching and mortality worldwide. The resulting 2014–17 database (Supplementary Data 1, Supplementary Fig. 1b) followed the Donner et al. 2017[69] format, including categories for (1) measures of coral bleaching as coral cover bleached (%), number of coral colonies bleached (n) and total number of colonies surveyed (N), or both; and/or (2) measures of coral mortality as coral cover dead (%), number of coral colonies dead (n) and total number of colonies surveyed (N), or both; (3) observation date; (4) observation location, including latitude, longitude, and reef site name; (5) data source; and (6) survey method used. We converted percentage bleached (and mortality, where available) into categorical variables following the same protocol as ReefBase (http://www.reefbase.org). While this simple method of bleaching collation has limitations (most notably, no requirement for data on bleaching by taxon), it allows for consistency in collation over time and inclusion of reports from rapid and low-technology bleaching assessments, as well as those conducted by citizen scientists through multiple programs (e.g., CORDIO-EA, ReefCheck, BleachWatch[70–73]). Additionally, we provided a qualitative questionnaire through which contributors could provide additional information about each observational dataset.

The database was compiled from data submitted directly by individuals and extractions from large databases. CRW sent out multiple calls for coral reef survey data (including observations of the absence of bleaching and/or mortality) conducted during the Third Global Coral Bleaching Event. We sought out a wide range of potential collaborators through direct contacts, calls for data via the NOAA Coral-List listserv, published calls for data[74], and an appeal through the team who created the film *Chasing Coral*[5]. We also extracted data on coral bleaching and mortality (where available) covering June 2014–May 2017 from multiple regional/international databases/sources including Donner et al. 2017[69], Reef Check International, Atlantic and Gulf Rapid Reef Assessment, Coastal Oceans Research and Development–Indian Ocean (CORDIO) East Africa, NOAA's National Coral Reef Monitoring Program (NCRMP)[76], and Great Barrier Reef surveys[77]. After removing surveys that lacked information critical to this study, the remaining 15,066 bleaching and mortality surveys (Supplementary Data 1) spanned the period June 1, 2014–May 31, 2017.

As the multiple individuals who contributed data to this paper used a variety of methods, the work presented here can be considered an unweighted meta-analysis of surveys conducted by numerous

individuals, institutions, and organizations during the 2014–17 Third Global Coral Bleaching Event. The techniques used were all highly comparable, well-accepted field methods. Past comparisons among coral reef survey methods have demonstrated that while some biases exist among methods, most provide comparable results across similar types of observations, such as percent coral cover[78,79]. The percentage of colonies bleached was often higher than the percentage of cover bleached because (1) small colonies bleached more often than large colonies; and/or (2) both partially- and fully-bleached colonies were counted as bleached in some survey methodologies. However, previous statistical comparison of the two methods found no significant differences when assessing the extent of coral bleaching or mortality[5]. Additionally, aggregating the data into broad categories minimized the need for precise estimates of bleaching. Therefore, we assumed that the different observation methods provide comparable results for this meta-analysis.

In our data, many surveys were conducted to meet multiple monitoring objectives, and the timing did not necessarily coincide with peak or even active local bleaching (e.g., established monitoring programs with pre-scheduled survey times)[80]. Conversely, some surveys may have been timed to coincide with predicted thermal stress events, likely informed by bleaching alerts based on CRW satellite products. In other cases, visits to particularly difficult-to-access survey locations may have been undertaken for similar reasons. However, because heat stress is an explicit covariate in our model, any such differential propensities for reefs to be surveyed would not introduce biases in our estimated thresholds; in this sense, our approach is more robust to such biases than raw reports of bleaching or mortality event frequency in survey data sets. Nevertheless, we do acknowledge that, if the actual existence of bleaching made a reef more likely to be surveyed than another reef that was unbleached but experiencing similar levels of heat stress, this could have led to overestimates of bleaching probability (as in all prior analyses of which we were aware).

## Data handling
We defined a unique bleaching observation (survey) as one with a unique combination of geographic coordinates, date, and depth or depth range. Each unique bleaching observation was assigned a data ID and included as a single, independent observation in the database. Multiple observations (quadrats or transects) taken at any reef site within a few days at the same depth range ($\pm 1$ m) were combined into a single survey, with bleaching by mean percent cover data or proportion of the number of colonies surveyed included in the database as the relevant observation. If percent bleached or mortality by individual colony data were provided, total coral area bleached was calculated by multiplying percent bleached or mortality of individual colonies by the size dimensions of each colony and dividing that number by the total area of all coral colonies at each site. Each survey was categorized by bleaching year and ocean basin based on coordinate boundaries: Asia-Pacific (AP, 100°E to 100°W crossing the anti-meridian), Caribbean/Atlantic (CA, 100°W to 20°E crossing the prime meridian), and Indian Ocean/Middle East (IM, 20°E to 100°E).

Because heat stress often reduces zooxanthellae (family Symbiodiniaceae) densities before bleaching becomes visible[81], coral colonies showing any degree of bleaching, from pale and partially bleached to fully bleached colonies, were identified as "bleached". Similarly, we included partial and complete mortality under "mortality", as it either indicated a heat stress response resulting in mortality due to bleaching or other, often heat-related, diseases. Therefore, partial and complete bleaching, and partial and complete mortality were combined as observations of bleaching and mortality, respectively.

As heat stress is not the only cause of bleaching and/or mortality of corals, some observations of bleaching or mortality may have been caused by other stress or predation. For this study, we used only bleaching observations that took place after satellite-based heat stress

had begun, and we used mortality data recorded after the peak of the heat stress event (i.e., the bleaching year maximum DHW) within a pixel or change in cover between repeat surveys during June 2014–May 2017. Bleaching was compared against either the heat stress at the time of the observation or that year's maximum heat stress if the observation was after the peak; mortality data were compared against that year's maximum heat stress. A prior analysis of reefs showed that 6% bleaching and 4% recent mortality (within the past year) normally existed as a background level during surveys in years lacking any major disturbance[82]. As previously established[5], a value of approximately twice these (10%) was used to define a threshold for moderate levels of bleaching and mortality. Severe bleaching was identified as being at least half of the population (>50%). Onset month of observed severe (>50%) bleaching in various regions across the globe is plotted in Supplementary Fig. 1d–f for each bleaching year. This revealed the general agreement of the bleaching sequence with that seen in 1998[10] and is generalized across the three bleaching years in Supplementary Fig. 1c and animated in Supplementary Video 1. Bleaching onset months during the Third Global Coral Bleaching Event can be compared with typical heat stress onset months found in CRW's Thermal History products[3,21].

Mortality data from the large databases only included corals known to have recently died (within the last year) or data that expert observers determined had recently died, most likely due to heat stress. In most cases, the actual cause of mortality was not known with complete certainty. Certain data contributors provided mortality data as a function of change over time (i.e., reduction in live coral cover since the previous survey). These data were listed as mortality present on the ending date of the reporting period.

Some monitoring programs reported percent bleaching or mortality in categories based on a range of values, and a few different sets of ranges were used across the contributing programs. These were adjusted for consistency by determining the ranges of the reported categories and then entering the mid-point value of that range into the database. Similarly, where data contributors provided ranges of percent bleached or mortality rather than single values, the mid-point values of the original ranges were used in the database.

## Data analyses
We fitted generalized linear mixed models (GLMMs) with binomial error structure using the library glmmTMB version 1.1.10 in version 4.4.2 of the R statistical software package[83,84]. We considered DHW, year, and ocean basin as the potential fixed effects, and the percent coral bleached or dead as the binomial response—i.e., whether the percentage of corals (area or number) in an individual reef survey exceeded the appropriate threshold (10% for moderate impact, 50% for severe impact) or not, for both bleaching and mortality. To check for spatial autocorrelation of residuals, we calculated between-site distances from latitude and longitude with the "geodesic" method, using the library "geodist" (version 0.1.0) in the software program R[84,85]; we computed semivariances using the function "Variogram" in the R package "nlme" (version 3.1.166)[86]. Models including only the fixed effects exhibited some evidence of spatial autocorrelation of residuals within the first several 1000 km of distance in the second and third bleaching years. This was particularly apparent within the first few 100 km in 2015–2016 for moderate or greater bleaching (Supplementary Fig. 10: note the steep initial trend in the blue circles of the top row, middle panel), and in 2016–2017 for moderate or greater bleaching (Supplementary Fig. 10: note the similar increase in the first few 100 km in the top row, right panel) and severe bleaching (Supplementary Fig. 11: note the similar increasing trend in the top row, right panel).

Spatial autocorrelation for mortality was also apparent in 2015–2016 and 2016–2017 (Supplementary Fig. 12 top-middle and top-right panels). Consequently, for all subsequent analyses, we added a

random effect of "Regional Virtual Station" which, for all bleaching observations, pulls together all surveys found within the boundaries of each of CRW's 214 Regional Virtual Stations[21,87]. Once these effects were included, there was much less apparent autocorrelation in either the residuals or the random effect estimates themselves (Supplementary Figs. 10–12, middle and bottom rows), and models including these random effects exhibited about 2000 AIC units better fit than the models omitting them. So, we used these mixed-effects models for further inference.

To assess the absolute performance of the best-fitting model (in addition to the relative performance of alternative models assessed by AIC), we calculated each model's prediction error: the proportion of events correctly predicted by the model (e.g., a correct bleaching prediction would be where the predicted probability of bleaching was >0.5, and bleaching occurred). We calculated prediction error both for the full model (including the random effects of Regional Virtual Station) and for the fixed effects component of the models only (omitting the random effects), analogous to "conditional" and "marginal" $R^2$ values from mixed-effects OLS regression. Previous bleaching studies most commonly reported prediction error, facilitating comparability of our models' performance with those used in earlier work[7,30,41,42].

To evaluate the evidence for the different possible effects and interactions among temperature, ocean basin, and bleaching year, we used Akaike Information Criterion (AIC) to compare the fit of models including full 3-way interactions (among DHW, bleaching year, and basin), with models removing all possible combinations of interactions involving bleaching year and basin. We wanted to consider models that had the same baseline level of bleaching at DHW = 0 °C-weeks but still differed in their sensitivity to heat stress. This corresponded to models where interactions between DHW and bleaching year, or ocean basin, or both were retained, but there was no "main effect" (i.e., no difference at the intercept where DHW = 0 °C-weeks) of year or basin, or no two-way interaction between year and basin.

Despite the voluminous observations in the 2014–17 bleaching database (15,066 survey data points), the uneven spatial distribution of sampling (Supplementary Fig. 1b) yielded direct, observational data from only 6.6% (3592 pixels) of the total 53,997 reef-containing pixels worldwide[21]. To predict the probabilities of >10% (moderate or greater) bleaching and mortality for all reef-containing pixels globally, we projected the statistically-modeled bleaching and -mortality thresholds for each basin and year from data on the maximum satellite-measured heat stress (DHW) at each reef-containing pixel.

We sought to predict the extent of bleaching and mortality globally for this event in a way that would not be biased by a propensity for surveys to have been conducted disproportionately on reefs experiencing particularly high heat stress. To do this, we used our fitted bleaching and mortality thresholds, together with maximum DHW values from each reef-containing pixel and for each of the three bleaching years of the Third Global Coral Bleaching Event, to predict the probability of moderate or greater bleaching and moderate or greater mortality in each bleaching year globally. Specifically, we used the estimated coefficients from the relevant GLMM (moderate or greater bleaching and moderate or greater mortality) for the effects of DHW, bleaching year, ocean basin, and associated interaction terms. We excluded the random effects of Regional Virtual Station ID since there were reef-containing pixels in Regional Virtual Stations from which we did not have observations.

Reef locations were assigned to one of 22 GCBE3 regions (Supplementary Table 3). Within each region, the probabilities were summed across reef-containing pixels for each of the bleaching years, yielding a measure of the predicted extent of bleaching (area of colored diamonds) and mortality (area within the sub-diamonds) for each period. Regions were designated as either Northern or Southern Hemisphere for the purpose of display, in which the predicted extent of bleaching was represented, with the color of each diamond

reflecting the maximum DHW during that bleaching year. Areas for each 12-month period (solid diamonds and enclosed dashed lines) were determined using fitted probabilities derived from the fixed effects of the best-fitting models of moderate or greater bleaching and moderate or greater mortality due to satellite-derived heat stress (Supplementary Fig. 3, Supplementary Fig. 5), summed across reef-containing pixels (i.e., two pixels with 30% probability of bleaching contribute the same as one pixel with a 60% probability of bleaching). The total reef area reflects the number of reef-containing pixels in each GCBE3 region (Fig. 4 inset). Summaries of all GCBE3 regional values are reported in Supplementary Table 3.

Comparisons of satellite-derived heat stress at each of the 214 NOAA CRW Satellite Regional Virtual Stations[21] to in-water observations of bleaching and mortality yielded very few mismatches between satellite-measured heat stress and observed response in corals (<5% of all observations). Cases of high DHW values (≥8 °C-weeks) with less than 10% bleaching or mortality, and those with no measured heat stress but more than 50% bleaching or 10% mortality in individual surveys were scattered in space and time and showed no geographic pattern. While the Third Global Coral Bleaching Event had a worldwide scope, there are many factors that confound relationships between heat stress and coral bleaching and mortality. These include poorly-timed observations that missed bleaching and mortality related to heat stress[80], and variations in bleaching and mortality with depth, light penetration, or habitat. In some sites, greater water depths have provided a layer of insulation from thermal events[88]. However, the potential for deep reefs to serve as refugia is uncertain and varies greatly[89–93]. The potential impact of depth on our analyses is described in the next section.

## Model selection

Model selection for moderate or greater bleaching (>10% of corals) indicated that the best-fitting model was one with all main effects and interactions, except for an interaction between ocean basin and bleaching year (i.e., baseline bleaching when DHW = 0 °C-weeks varied among years, and among basins, but year effects were consistent across basins and vice-versa). However, the retained three-way interaction involving DHW implies that the sensitivity of bleaching to temperature varied differently among basins in different years (see Supplementary Table 4 for model selection; Supplementary Fig. 3 for fitted thresholds of best-fitting model). For severe bleaching (>50% of coral bleached), the best-fitting model implied that all bleaching years and ocean basins had a common baseline bleaching value at DHW = 0 °C-weeks (i.e., no main effects of basin or year, and no interaction between basin and year), but that the severe bleaching thresholds did vary in different ways among years and basins, as in the "moderate bleaching" analysis (see Supplementary Table 5 for model selection; Supplementary Fig. 4 for best-fitting model).

For moderate or greater mortality (>10% of corals), the best-fitting model included all main effects and interactions involving DHW, ocean basin, and bleaching year (see Supplementary Table 6 for model selection; Supplementary Fig. 5 for best-fitting model). For Fig. 2 we fitted models including only fixed effects and interactions for DHW and bleaching year (for panel a), and DHW and basin (for panel b), to assist visualization of overall differences in thresholds among years and among basins. However, it is important to note that the best model for these data included interactions among all three fixed effects; those fitted relationships were plotted in Supplementary Fig. 3.

Finally, finer-scale patterns of bleaching or mortality in nearshore reefs may have resulted from localized heating, cooling, or shading at spatial resolutions not detectable by the heat stress metric used in this study. To test if depth had an impact on the models, sensitivity analyses incorporated depth as a covariate on the observations where depth was recorded. Observation depths were not consistent among basins with those from the Caribbean tending to be somewhat deeper, on average, than those from the other two ocean basins

(Supplementary Fig. 9). The sensitivity analysis showed that the basin and inter-annual variations in moderate bleaching, severe bleaching, and moderate mortality thresholds were robust to these differences (Supplementary Figs 10–12), therefore we believe the analyses that included the full dataset and omitted depth provided a stronger representation of the bleaching and mortality relationships with heat stress across the three basins.

## Ethics and inclusion

This work relied on both satellite and field observations. We sought out all available field observations of coral bleaching and mortality from any coral reef-containing countries. At least one, and often multiple, members of each research team contributing observations participated in the preparation of data for this analysis and thus were included as authors and approved the manuscript. This offer was made at the time the data were contributed. All researchers involved in field surveys are identified in the observational database.

## Reporting summary

Further information on research design is available in the Nature Portfolio Reporting Summary linked to this article.

## Data availability

Sea surface temperature data and heat stress metrics are available from https://coralreefwatch.noaa.gov/product/5km/index.php and archived at the National Centers for Environmental Information at https://doi.org/10.25921/6jgr-pt28. All other data generated or analyzed during this study are included in this published article and its supplementary information files. Data are provided with this paper as follows: The site-level metadata (e.g., depth, latitude, longitude) and bleaching and mortality observations used in this study are available as Supplementary Data 1. The remote sensing data, all fitted model objects, and the code necessary to regenerate fitted model objects, and to reproduce all figures, tables, and other results in this paper, are available on Zenodo (https://doi.org/10.5281/zenodo.15114357, https://zenodo.org/records/15114357).

## Code availability

Code to reproduce the analyses presented in this paper, data files needed for such reproduction, and files containing all fitted model objects used in the final production of bleaching and mortality results may be found on Zenodo at https://zenodo.org/records/15114357.

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

## Acknowledgements

The U.S. National Oceanic and Atmospheric Administration's (NOAA) CRW program and National Coral Reef Monitoring Program were sup-ported by funding from the NOAA Coral Reef Conservation Program and Ocean Remote Sensing Program. University of Maryland and ReefSense personnel were fully supported by NOAA grant NA19NES4320002 (Cooperative Institute for Satellite Earth System Studies) at the Uni-versity of Maryland/Earth System Science Interdisciplinary Center, and by the Professional, Scientific, and Technical Services Program (Pro-Tech)-Satellite contract with Global Science & Technology, Inc. SFH was partially supported by Australian Research Council grant DP230102986. Bleaching analysis work by SRC was supported by the Australian Research Council (CE140100020) and the Smithsonian Tropical Research Institute. Part of this work was performed and funded under ST133017CQ0050_1332KP22FNEED0042. Additional funding for data collation and analysis provided by Vulcan Inc. We also acknowledge, with gratitude, the myriad funding sources that supported the collection of data that enabled this analysis. The scientific results and conclusions, as well as any views or opinions expressed herein, are those of the author(s) and do not necessarily reflect the views of NOAA or the Department of Commerce. Figure 4 was inspired by a National Geo-graphic infographic[94]. We thank J. Moneghetti for assistance with sta-tistical programming. Thank you to the hundreds of individual data collectors from organizations such as Reef Check, CORDIO, ILTER/PELD (Brazil), and NOAA around the world who contributed to this dataset. In particular, we thank the hundreds of volunteers and the following team leaders from almost 30 Reef Check chapters, who organized teams, carried out surveys and provided significant datasets for these analyses: Australia: Jennifer Loder, Jodi Salmod, Bahamas: Lourene Jones, Mon-ique Curtis, Tom McFeely, Brunei: Sheikh Al-Idrus Nikman, Colombia: Phanor Montoya, Egypt: Nina Milton, Mohammad Kotb, Moshira Hassan, Florida: Nikole Ordway, France (Pacific): Jean Pascale Quod, Matthieu Petit, Denis Schneider, Harold Cambert, France (Atlantic): Remi Garnier, Mathilde Facon, Grenada: Katlyn Treiber-Vajda, Haiti: Erika Pierre Louis, Indonesia: Delphine Robbe, Gianfranco Rossi, Meike Huhn, Andrew Taylor, Nyoman Sugiarto, Iran: Mohammad Ghavasi, Japan: Yasuaki Miamoto, J Harukawa, Satoshi Nojima, Megumu Tsuchikawa, Maldives: Jean-Luc Solandt, Matthias Hammer, Catherine Edsell (Biosphere Expeditions), Malaysia: Julian Hyde, Sue Yee Chen, Alvin Chelliah, Netherlands Antilles: Marjo van den Brulck, Oman: Jean-Luc Solandt, Matthias Hammer, Catherine Edsell (Biosphere Expeditions), Philippines: Vanessa Vergara, Carina Escudero, Xavier Verdadero, Analies Andringa, Scott Countryman, Colin Lock, Puerto Rico: Joel Melendez, Carolina Aragones, St Kitts/Nevis: James Hewlett, Sao Tome/Principe: An Bollen, Thailand: Nathan Cook, Suchana Chavanich, Timor-Leste: Jenny House, Tobago: Lanya Fanovich, Taiwan: Kah-Leng Cherh; and Healthy Reefs for Healthy People Initiative country coordinators who organized teams, carried out surveys and provided significant datasets for these analyses: Belize: Nicole Craig, Guatemala: Ana Giró Petersen. We are grateful for the work of myriad other scientists and non-scientists who contributed data used herein. We acknowledge the Traditional Custodians of coral regions across the tropics from which datasets used here were collected and recognize these First Nations peoples as among the world's earliest reef scientists.

## Author contributions

The study was conceptualized by C.M.E., who also wrote the first draft of the paper. C.M.E., S.F.H., and S.R.C. jointly led revision of the manuscript. D.A.D. coordinated data compilation in conjunction with J.L.D. and A.M.G. S.R.C. designed and led the statistical analysis. G.L., W.J.S., E.F.G., and S.F.H. provided data on heat stress. Large-scale coral bleaching and mortality datasets were provided by A.H.B., N.C., C.S.C., S.D.D., J.G., M.G.R., M.G., H.B.H., G.H., O.H.G., A.S.H., M.O.H., T.P.H., M.E.J., J.T.K., T.K., J.M., A.I.M.-C., D.O.O., M.S.P., A.R.-S., C.L.R., A.S., J.S., A.T., G.T., T.S.V., C.S.W., and S.W. Great Barrier Reef aerial bleaching surveys were undertaken by T.P.H. and J.T.K. All other authors contributed to data collection and multiple drafts of the manuscript.

## Competing interests

The authors declare no competing interests.

## Additional information

C. Mark Eakin [1,2] ✉, Scott F. Heron [1,3,4] ✉, Sean R. Connolly [3,5] ✉, Denise A. Devotta [1,6], Gang Liu [1], Erick F. Geiger [1,6], Jacqueline L. De La Cour [1,7], Andrea M. Gomez [8], William J. Skirving [1,9,10], Andrew H. Baird [3,4], Neal E. Cantin [11], Courtney S. Couch [12,13], Simon D. Donner [14], James Gilmour [11,15], Manuel Gonzalez-Rivero [11], Mishal Gudka [16], Hugo B. Harrison [17], Gregor Hodgson [18,19], Ove Hoegh-Guldberg [20], Andrew S. Hoey [4], Mia O. Hoogenboom [3,4], Terry P. Hughes [4], Meaghan E. Johnson [21], James T. Kerry [4], Tadashi Kimura [22], Jennifer Mihaly [19], Aarón Israel Muñiz-Castillo [23], David O. Obura [16], Morgan S. Pratchett [3,4], Andrea Rivera-Sosa [23], Claire L. Ross [15,24], Jennifer Stein [25], Angus Thompson [11], Gergely Torda [4], T. Shay Viehman [26], Cory S. Walter [27], Shaun Wilson [11,24], Benjamin L. Marsh [1,9,146], Blake L. Spady [1,6], Noel Dyer [28], Thomas C. Adam [29], Pedro Alcolado [30,147], Mahsa Alidoostsalimi [31], Parisa Alidoostsalimi [32], Lorenzo Alvarez-Filip [33], Mariana Álvarez-Noriega [4], Jesús Ernesto Arias-González [23], Keisha D. Bahr [34], Peter Barnes [35], José Enrique Barraza Sandoval [36], Julia K. Baum [37,38], Andrew G. Bauman [39], Maria Beger [40,41], Kathryn Berry [42], Pia Bessell-Browne [43], Lionel Bigot [44], Victor Bonito [45], Ole B. Brodnicke [46], David Burdick [47], Deron E. Burkepile [48], April J. Burt [49], John A. Burt [50], Ian R. Butler [4,51], Jamie M. Caldwell [52], Yannick Chancerelle [53], Chaolun Allen Chen [54], Kah-Leng Cherh [55], Michael J. Childress [56], Darren J. Coker [57], Bryan Costa [26], Georgia Coward [58], M. James C. Crabbe [59], Thomas Dallison [60], Steven Dalton [61], Thomas M. DeCarlo [62], Crawford Drury [38], Ian Drysdale [63], Clinton B. Edwards [64], Linda Eggertsen [38], Eylem Elma [65], Rosmin S. Ennis [66], Richard D. Evans [15,24], Gal Eyal [20,67], Douglas Fenner [68], Baruch Figueroa-Zavala [69], Jay Fisch [70], Michael D. Fox [57], Elena Gadoutsis [71], Antoine Gilbert [72,73], Andrew R. Halford [74], Tom Heintz [72], James Hewlett [75], Jean-Paul A. Hobbs [76], Whitney C. Hoot [77], Peter Houk [47], Lyza Johnston [78], Michelle A. Johnston [79], Hajime Kayanne [80], Emma V. Kennedy [11], Ruy Kenji Papa de Kikuchi [81], Ulrike Kloiber [82], Haruko Koike [83], K. Lindsey Kramer [84], Chao-Yang Kuo [4,54,55], Judith Lang [85], Alice Lawrence [86], Abigail Leadbeater [87], Zelinda M. A. N. Leão [81], Jen Nie Lee [88], Cynthia Lewis [89], Diego Lirman [70], Guilherme Ortigara Longo [90], Chancey MacDonald [6,91], Jennie Mallela [92], Sangeeta Mangubhai [93], Isabel Marques da Silva [94], Christophe Mason-Parker [95], Vanessa McDonough [96], Melanie McField [97], Thayná Mello [90], Celine Miternique-Agathe [98], Mouchtadi Madi [99], Stephan Moldzio [100], Alison A. Monroe [101], Monica Montefalcone [102], Kevin S. Moses [103], Pargol G. Mostafavi [104], Rodrigo Leão de Moura [105], Chathurika S. Munasinghe [106], Jelvas Mwaura [107], Takashi Nakamura [108], Jean-Benoit Nicet [109], Marissa F. Nuttall [79], Marilia D. M. Oliveira [81], Hazel A. Oxenford [110], John M. Pandolfi [20], Vardhan Patankar [111], Denise Perez [112], Nishan Perera [113], Derta Prabuning [114], William Precht [115], K. Diraviya Raj [116], James D. Reimer [108], Laura E. Richardson [86], Randi Rotjan [117], Nicole Ryan [118], Rod Salm [119], Stuart A. Sandin [64], Stephanie Schopmeyer [120], George Shedrawi [74,121], Mohammad Reza Shokri [122], Jennifer E. Smith [64], Kylie Smith [56,123], Struan R. Smith [124], Tyler B. Smith [66], Brigitte Sommer [125], Melina Soto [126], Richard Suckoo [127], Helen Sykes [128], Kelley Anderson Tagarino [129], Marianne Teoh [130], Minh Quang Thai [131], Tai Chong Toh [132], Alex Tredinnick [133], Alex Tso [14], Harriet Tyley [134], Ali M. Ussi [135], Bernardo Vargas-Angel [136], Christian Vaterlaus [137], Mark J. A. Vermeij [138], Si Tuan Vo [131], Christian R. Voolstra [139], Hin Boo Wee [140,141], Bradley A. Weiler [70], Dana E. Williams [142], Saleh A. S. Yahya [143], Thamasak Yeemin [144], Maren Ziegler [145] & Derek P. Manzello [1]

[1]Coral Reef Watch, Center for Satellite Applications and Research, U.S. National Oceanic and Atmospheric Administration, College Park, MD, USA. [2]Corals and Climate, Silver Spring, MD, USA. [3]College of Science and Engineering, James Cook University, Townsville, QLD, Australia. [4]Australian Research Council

Centre of Excellence for Coral Reef Studies, James Cook University, Townsville, QLD, Australia. [5]Smithsonian Tropical Research Institute, Balboa, Ancón, Panama. [6]Global Science & Technology LLC, Greenbelt, MD, USA. [7]Earth System Science Interdisciplinary Center, Cooperative Institute for Satellite Earth System Studies, University of Maryland, College Park, MD, USA. [8]Greater Atlantic Regional Office, U.S. National Oceanic and Atmospheric Administration Fisheries, Gloucester, MA, USA. [9]ReefSense, Cranbrook, QLD, Australia. [10]University of Reading, Reading, UK. [11]Australian Institute of Marine Science, PMB 3, Townsville, QLD, Australia. [12]Cooperative Institute for Marine and Atmospheric Research, University of Hawaii at Manoa, Honolulu, HI, USA. [13]Pacific Islands Fisheries Science Center, National Marine Fisheries Service, National Oceanic and Atmospheric Administration, Honolulu, HI, USA. [14]Institute for Resources, Environment and Sustainability, Department of Geography, University of British Columbia, Vancouver, BC, Canada. [15]Oceans Institute, University of Western Australia, Crawley, WA, Australia. [16]CORDIO East Africa, Mombasa, Kenya. [17]School of Biological Sciences, University of Bristol, Bristol, UK. [18]Angelo King Center for Research & Environmental Management, Silliman University, Dumaguete, Philippines. [19]Reef Check Foundation, Westlake Village, CA, USA. [20]School of The Environment, The University of Queensland, Brisbane, QLD, Australia. [21]National Parks of Eastern North Carolina, Manteo, NC, USA. [22]Palau International Coral Reef Center, Koror, Palau. [23]Laboratorio de Ecología de Ecosistemas de Arrecifes Coralinos, Departamento de Recursos del Mar, Centro de Investigación y de Estudios Avanzados del I.P.N., Mérida, Yucatán, Mexico. [24]Department of Biodiversity, Conservation and Attractions. Kensington, Perth, WA, Australia. [25]Fish and Wildlife Research Institute, Marathon, FL, USA. [26]National Centers for Coastal Ocean Science, U.S. National Oceanic and Atmospheric Administration, Santa Barbara, CA, USA. [27]International Center for Coral Reef Research and Restoration, Mote Marine Laboratory, Summerland Key, FL, USA. [28]Office of Coast Survey, U.S. National Oceanic and Atmospheric Administration, Silver Spring, MD, USA. [29]Marine Science Institute, Moorea Coral Reef LTER, University of California Santa Barbara, Santa Barbara, CA, USA. [30]Instituto de Oceanologia, Havana, Cuba. [31]School of Geography, Earth and Atmospheric Sciences, Faculty of Science, The University of Melbourne, Melbourne, VIC, Australia. [32]Department of Marine Biology, Science and Research Branch, Azad University, Tehran, Iran. [33]Biodiversity and Reef Conservation Laboratory, Unidad Académica de Sistemas Arrecifales, Instituto de Ciencias del Mar y Limnología, Universidad Nacional Autónoma de México, Puerto Morelos, Quintana Roo, Mexico. [34]Texas A&M University - Corpus Christi, Corpus Christi, TX, USA. [35]Ningaloo Marine Park, Parks and Wildlife Service - Exmouth District, Department of Biodiversity, Conservation and Attractions, Exmouth, WA, Australia. [36]Ministerio de Medio Ambiente y Recursos Naturales, Dirección de Ecosistemas y Biodiversidad, San Salvador, El Salvador. [37]Department of Biology, University of Victoria, Victoria, BC, Canada. [38]Hawaii Institute of Marine Biology, University of Hawaii, Kāne'ohe, HI, USA. [39]National Coral Reef Institute, Halmos College of Arts and Sciences, Department of Marine and Environmental Science, Nova Southeastern University, Dania Beach, FL, USA. [40]School of Biology, Faculty of Biological Sciences, University of Leeds, Leeds, UK. [41]Centre for Biodiversity and Conservation Science, School of the Environment, University of Queensland, Brisbane, QLD, Australia. [42]Fisheries and Oceans Canada, Sidney, BC, Canada. [43]CSIRO, Castray Esplanade, Battery Point, TAS, Australia. [44]Entropie (IRD, UR, CNRS, UNC, IFREMER), Labex CORAIL, IRD-University of La Réunion, Saint Denis Cedex 9, France. [45]Reef Explorer Fiji, Votua Village, Nadroga, Fiji. [46]Offshore Biodiversity and Ecology, DHI A/S, Hørsholm, Denmark. [47]University of Guam Marine Laboratory, Mangilao, GU, USA. [48]Department of Ecology, Evolution, & Marine Biology, Moorea Coral Reef LTER, University of California Santa Barbara, Santa Barbara, CA, USA. [49]The Seychelles Islands Foundation, Mont Fleuri, Mahe, VIC, Seychelles. [50]Mubadala ACCESS Center & Center for Genomics and Systems Biology, New York University Abu Dhabi, PO Box, Abu Dhabi, United Arab Emirates. [51]CoraLogic Environmental Consulting, Cook, ACT, Australia. [52]High Meadows Environmental Institute, Princeton University, Princeton, NJ, USA. [53]Centre de Recherches Insulaires et Observatoire de l'Environnement/SNO Corail, Papetoai, Moorea Island, French Polynesia, France. [54]Biodiversity Research Center, Academia Sinica, Nangang, Taipei, Taiwan, ROC. [55]Taiwan Environmental Information Association, New Taipei City, Taiwan, ROC. [56]Department of Biological Sciences, Clemson University, Clemson, SC, USA. [57]Red Sea Research Center, Division of Biological and Environmental Science and Engineering, King Abdullah University of Science and Technology, Thuwal, Saudi Arabia. [58]Department of Marine and Wildlife Resources, American Samoa Coral Reef Advisory Group, Pago Pago, AS, USA. [59]Wolfson College, Oxford University, Oxford, UK. [60]Blue Pangolin Consulting Ltd, London, UK. [61]Department of Primary Industries, NSW Fisheries, Coffs Harbour, NSW, Australia. [62]Department of Earth and Environmental Sciences, Tulane University, New Orleans, LA, USA. [63]Healthy Reefs for Healthy People Initiative, West End, Roatan, Honduras. [64]Scripps Institution of Oceanography, UC San Diego, La Jolla, CA, USA. [65]Department of Biology, University of Oxford, Oxford, UK. [66]Center for Marine and Environmental Studies, University of the Virgin Islands, St. Thomas, USVI, USA. [67]The Mina & Everard Goodman Faculty of Life Sciences, Bar-Ilan University, Ramat Gan, Israel. [68]Coral Reef Consulting, Pago Pago, AS, USA. [69]Centro Ecológico Akumal, Akumal, Mexico. [70]Rosenstiel School of Marine, Atmospheric and Earth Science, University of Miami, Miami, FL, USA. [71]Save Our Seas Foundation - D'Arros Research Centre, Genève, Switzerland. [72]Ginger-Soproner, BP3583 Nouméa cedex, New Caledonia, France. [73]Direction des Ressources Marines, Fare Ute, Papeete, Tahiti, French Polynesia, France. [74]Pacific Community (SPC), Nouméa, New Caledonia, France. [75]Finger Lakes Community College, Canandaigua, NY, USA. [76]School of Biological Sciences, University of Queensland, Brisbane, QLD, Australia. [77]Guam Coral Reef Initiative, Bureau of Statistics and Plans, Hagatna, Guam, USA. [78]Johnston Applied Marine Sciences, Saipan, CNMI, USA. [79]NOAA Flower Garden Banks National Marine Sanctuary, Galveston, TX, USA. [80]The University of Tokyo, Hongo, Tokyo, Japan. [81]Department of Oceanography - Federal University of Bahia, RECOR - Coral Reefs and Global Change Lab, National Institute of Science and Technology for the Tropical Marine Environments (INCT AmbTropic), Salvador, Bahia, Brazil. [82]Chumbe Island Coral Park, Zanzibar, Tanzania. [83]Fishery Solutions LLC, Yokohama City, Kanagawa, Japan. [84]State of Hawai'i Division of Aquatic Resources, Kailua-Kona, HI, USA. [85]Atlantic and Gulf Rapid Reef Assessment (AGRRA), Big Pine Key, FL, USA. [86]School of Ocean Sciences, Bangor University, Anglesey, UK. [87]Blue Ventures, The Old Library, Bristol, UK. [88]Faculty of Science and Marine Environment, Universiti Malaysia Terengganu, Kuala Nerus, Terengganu, Malaysia. [89]Department of Biology, Florida International University, Miami, FL, USA. [90]Marine Ecology Laboratory, Department of Oceanography and Limnology, Universidade Federal do Rio Grande do Norte, Natal, RN, Brazil. [91]California Academy of Sciences, San Francisco, CA, USA. [92]Research School of Biology, Australian National University, Canberra, ACT, Australia. [93]Talanoa Consulting, Suva, Fiji. [94]Lúrio University, Nampula, Cabo Delgado, Mozambique. [95]Marine Conservation Society Seychelles, Mahé, Victoria, Seychelles. [96]Biscayne National Park, Homestead, FL, USA. [97]Healthy Reefs Initiative/Smithsonian Institution, Ft. Pierce, FL, USA. [98]Reef Conservation, Pereybere, Cap Malheureux, Mauritius. [99]Moheli Marine Park, Nioumachoua, Comoros. [100]Green Corals, Rischauer Moor 1, Braunschweig, Germany. [101]Department of Life Sciences, Marine Genomics Laboratory, Texas A&M University Corpus Christi, Corpus Christi, TX, USA. [102]DiSTAV, University of Genova, Corso Europa 26, Genova, Italy. [103]Zoological Survey of India, Marine Biology Regional Center, Chennai, India. [104]Department of Marine Sciences, Faculty of Natural resources and Environment, Science and Research Branch, Islamic Azad University (I.A.U), Tehran, Iran. [105]Instituto de Biologia and SAGE/COPPE, Universidade Federal do Rio de Janeiro, Rio de Janeiro, Brazil. [106]Department of Zoology, Faculty of Science, University of Peradeniya, Peradeniya, Sri Lanka. [107]Kenya Marine and Fisheries Research Institute, Mombasa, Kenya. [108]Faculty of Science, University of the Ryukyus, Okinawa, Japan. [109]MAREX Saint-Leu Reunion, Saint-Leu, France. [110]CERMES, University of the West Indies, Cave Hill, Saint Michael, Barbados. [111]GVI Travel II Limited t/a GVI Experiences, Exeter, England. [112]The Nature Conservancy Caribbean Division, San Juan, PR, USA. [113]Blue Resources Trust, Colombo, Sri Lanka. [114]Reef Check Indonesia Foundation, Denpasar, Bali, Indonesia. [115]Coastal and Marine Sciences, Bio-Tech Consulting, Miami, FL, USA. [116]Suganthi Devadason Marine Research Institute, Tuticorin, Tamil Nadu, India. [117]Department of Biology, Boston University, Boston, MA, USA. [118]Australian Institute of Marine Science, Indian Ocean Marine Research Centre, Crawley, WA, Australia. [119]The Nature Conservancy (Retired), Kailua, HI 96734; Science Advisory Board, Coral Triangle Center, Bali, Indonesia. [120]Fish and

Wildlife Research Institute, St. Petersburg, FL, USA. [121]Australian Centre for Ocean Resources and Security, University of Wollongong, Wollongong, NSW, Australia. [122]Faculty of Life Sciences and Biotechnology, Shahid Beheshti University, Tehran, Iran. [123]I.CARE, Islamorada, FL, USA. [124]Natural History Museum, Bermuda Aquarium, Museum and Zoo, Flatt's, Bermuda. [125]School of Life and Environmental Science, The University of Sydney, Sydney, NSW, Australia. [126]Healthy Reefs for Healthy People Initiative, Puerto Morelos, Quintana Roo, Mexico. [127]Coastal Zone Management Unit, Barbados, and University of the West Indies, Saint Michael, Barbados. [128]Marine Ecology Consulting, Suva, Fiji. [129]University of Hawaii Sea Grant College Program, Honolulu, HI, USA. [130]Fauna & Flora International, Cambridge, UK. [131]Institute of Oceanography, Vietnam Academy of Science and Technology, Nha Trang, Khanh Hoa, Vietnam. [132]Tropical Marine Science Institute, National University of Singapore, Singapore, Singapore. [133]MISE Lab, The University of Queensland, Brisbane, QLD, Australia. [134]Coral Cay Conservation, Tongham, Surrey, UK. [135]State University of Zanzibar, Zanzibar, Tanzania. [136]NOAA Restoration Center, Charleston, SC, USA. [137]marinecultures.org, Zanzibar, Tanzania. [138]Carmabi Foundation/ University of Amsterdam, Willemstad, Curaçao. [139]Department of Biology, University of Konstanz, Konstanz, Germany. [140]Centre for Tropical Climate Change System, Institute of Climate Change, Universiti Kebangsaan Malaysia, Bangi, Selangor, Malaysia. [141]Institute of Oceanography and Environment (INOS), Universiti Malaysia Terengganu, Kuala Nerus, Terengganu, Malaysia. [142]Cooperative Institute for Marine and Atmospheric Studies, University of Miami, Miami, FL, USA. [143]Institute of Marine Sciences, University of Dar es Salaam, Zanzibar, Tanzania. [144]Marine Biodiversity Research Group, Department of Biology, Faculty of Science, Ramkhamhaeng University, Bangkok, Thailand. [145]Department of Holobiont Biology, Justus Liebig University Giessen, Giessen, Germany. [146]Deceased: Benjamin L. Marsh. [147]Deceased: Pedro Alcolado. ✉e-mail: corals.and.climate@gmail.com; scott.heron@jcu.edu.au; ConnollyS@si.edu

