## [Transparent Peer Review file · Nature Communications]

Severe and widespread coral reef damage during the 2014-2017 Global Coral Bleaching Event

Corresponding Author: Dr C. Mark Eakin

Version 0:

Reviewer comments:

Reviewer #1

(Remarks to the Author)

While I was not a Reviewer of the original submission of this manuscript, I have carefully read the previous reviewers' comments and the authors' responses to those comments.

In my view, the authors have constructively and adequately addressed the previous reviewer's critiques, and as a result the manuscript is now clearer, more rigorous, and more compelling.

Furthermore, I reviewed the revised manuscript – and did so prior to reading the response letter (to establish my own, unbiased assessment).

I found the manuscript to be very interesting, technically sound, and well-written (which is impressive, given both the scope and nuance of the underlying data). The authors impressive blending of global-scale in situ observations with remotely sensed data conveys a clear and important story about the severity and extent of the third global coral bleaching event and the spatiotemporal variance of this phenomenon. The revised version is solid.

Per the prior comments and response letter, I agree you should leave Figure 1 as is (re: alert levels) – as I agree that this metric is best suited for a broad audience – but I encourage the authors to include the DHW version in supplement as well, for the benefit of scientists working on this specific topic matter.

In my view, this MS is ready for publication following small editorial changes (see below) and any subsequent edits deemed necessary by R2.

Minor editorial comments:

Line 270-271: Unclear what “these” is referring to – I presume warming and coral disease? Please clarify.

Line 299: “and major losses on the GBR are expected for 2023–24”. Were incurred? Do we know yet?

Line 302: As written (and given the previous few sentences), the word “dominated” gives the uninformed reader the impression that Caribbean reefs now generally display high coral cover (underpinned by a few thermally tolerant taxa). But in reality, most Caribbean reefs have very low coral cover (and are not dominated by coral at all, but rather other groups like seaweed). Thus, change “dominated” to “characterized” (or similar) to avoid confusion.

Reviewer #2

(Remarks to the Author)

This manuscript has been extensively reviewed and revised by the authors at least two times in this journal by other reviewers. As a new reviewer to this process in the third round, I generally agree that the authors have done as best a job they can with the previous sets of reviews (including the responses to the latest round of reviews). At an earlier stage, I would suggest a revised focus or statistical approach, but that doesn't feel appropriate now. My major suggestion is on the framing – as Reviewer 1 notes, this bleaching event occurred over 10 years ago and another one (in 2024, named the Fourth Global Bleaching Event - so a major disturbance) was also purported as the most severe and widespread. Without a direct comparison between the 2014-2017 and the 2024 event, I don't think the authors can justify their title or conclusions – that this is “The Most Severe and Widespread Coral Reef Damage Ever Recorded”. I recommend acceptance pending a re-framing, i.e. that this is a 10-year review of the 2014-2017 event and that similar analyses of the 2024 event deserve similar comparison, at least of the temperature / DHW / satellite data (since gathering the monitoring data from so many different sources is likely what took 10 years!) I would also take the moment to appreciate the work that many NOAA scientists and technologists have done over many years to advance this work, and that I hope the NOAA datasets remain useful to scientists for years to come. In the current political climate, it's also worth noting that publishing this article quickly will be a record of the usefulness of NOAA's tracking and forecasting of coral bleaching events worldwide.

Major suggestions:

Title. Change to, “The 2014–17 Global Coral-Bleaching Event: A Record of Severe and Widespread Coral Reef Damage”

Abstract. Identify Fourth Global Bleaching Event and how these findings can help inform continued predictions / understanding of bleaching events.

Personally I find the ‘GCBE’ acronyms to be distracting and not a common acronym, I would suggest removing and replacing with ‘bleaching events’ or ‘Third Global Bleaching Event’, etc.

Fig1 - Any chance to add a panel of NOAA Alerts for the 2024 event to put these panels in context with the most recent event? I'm definitely not asking for any addition or extra analysis of the monitoring data / bleaching simulations - just simply for the thermal context of these events.

Minor comments

L106 - Was this ‘consistent spatial evolution’ also what you observed during this event? Or did this bleaching event follow a different spatial expansion? I suggest replacing ‘evolution’ with ‘expansion’ to be more appropriate (it's not an evolution over many generations, per se)

Response to reviews

We thank the reviewers for their consideration of our manuscript, and we are grateful for the opportunity to submit this revised manuscript. We have responded to each of the comments provided as detailed below, in most cases leading to changes in the manuscript main text.

We specifically note:

- we have prioritised expediting the response to these review comments in acknowledgement that several co-authors are U.S. Federal government employees whose capacity to engage in a timely fashion may be affected in the near future
- some authors have been re-instated as their contact information had changed prior to the submission to *Nature Communications* (and following the earlier submission to *Nature*)
- the reviewers' statements, with the revisions/reframing undertaken based on the reviewers' comments, that the manuscript is "ready for publication" (R1) and to "recommend acceptance" (R2).

Reviewer 1

R1.1 While I was not a Reviewer of the original submission of this manuscript, I have carefully read the previous reviewers' comments and the authors' responses to those comments.

In my view, the authors have constructively and adequately addressed the previous reviewer's critiques, and as a result the manuscript is now clearer, more rigorous, and more compelling.

Response: We thank the reviewer for these comments on the efforts undertaken previously.

R1.2 Furthermore, I reviewed the revised manuscript – and did so prior to reading the response letter (to establish my own, unbiased assessment).

I found the manuscript to be very interesting, technically sound, and well-written (which is impressive, given both the scope and nuance of the underlying data). The authors impressive blending of global-scale in situ observations with remotely sensed data conveys a clear and important story about the severity and extent of the third global coral bleaching event and the spatiotemporal variance of this phenomenon. The revised version is solid.

Response: We are grateful for this support of the scope and quality of our submission.

R1.3 Per the prior comments and response letter, I agree you should leave Figure 1 as is (re: alert levels) – as I agree that this metric is best suited for a broad audience – but I encourage the authors to include the DHW version in supplement as well, for the benefit of scientists working on this specific topic matter.

Response: We have now included charts of maximum DHW for each of these three global coral bleaching events in the Extended Data (now ED Fig. 2).

R1.4 In my view, this MS is ready for publication following small editorial changes (see below) and any subsequent edits deemed necessary by R2.

Response: We thank the reviewer for this assessment.

Minor editorial comments:

R1.5 Line 270-271: Unclear what “these” is referring to – I presume warming and coral disease? Please clarify.

Response: We have replaced “these” with “bleaching- and disease-related mortality”.

*“Together with impacts from overfishing, land-based pollution, and sea urchin diseases, **bleaching- and disease-related mortality** have reduced Caribbean/Atlantic coral density and cover [44], particularly among acroporid corals [47], as well as diversity [48].” (lines 272-274)*

R1.6 Line 299: “and major losses on the GBR are expected for 2023–24”. Were incurred? Do we know yet?

Response: We have rephrased to clarify and to provide a reference (in a nutshell extensive bleaching was documented but effects on mortality and overall coral cover are not yet known). We have added information and a citation for what is known so far from the Great Barrier Reef:

*“For example, recovery of coral cover on the Great Barrier Reef was reported through 2022 following recent bleaching events [56]. While encouraging, this was driven by fast-growing *Acropora* corals that are highly susceptible to bleaching during subsequent heat stress, wave damage from storms and tropical cyclones, and crown-of-thorns starfish predation. This ‘recovery’ measured by cover may be masking a concomitant loss of coral diversity and associated functional traits [50]. Unfortunately, bleaching in 2022 ended that recovery trend [57] and **in 2023-24, initial analysis reported moderate or greater (>10%) bleaching of 73% of survey reefs on the Great Barrier Reef [58], with major losses expected.**” (lines 296-303)*

R1.7 Line 302: As written (and given the previous few sentences), the word “dominated” gives the uninformed reader the impression that Caribbean reefs now generally display high coral cover (underpinned by a few thermally tolerant taxa). But in reality, most Caribbean reefs have very low coral cover (and are not dominated by coral at all, but rather other groups like seaweed). Thus, change “dominated” to “characterized” (or similar) to avoid confusion.

Response: We concur with the reviewer’s suggestion and have replaced “dominated” with “characterized”.

Reviewer 2

R2.1 This manuscript has been extensively reviewed and revised by the authors at least two times in this journal by other reviewers. As a new reviewer to this process in the third round, I generally agree that the authors have done as best a job they can with the previous sets of reviews (including the responses to the latest round of reviews).

Response: We thank the reviewer for these comments.

R2.2 At an earlier stage, I would suggest a revised focus or statistical approach, but that doesn't feel appropriate now. My major suggestion is on the framing – as Reviewer 1 notes, this bleaching event occurred over 10 years ago and another one (in 2024, named the Fourth Global Bleaching Event - so a major disturbance) was also purported as the most severe and widespread. Without a direct comparison between the 2014-2017 and the 2024 event, I don't think the authors can justify their title or conclusions – that this is “The Most Severe and Widespread Coral Reef Damage Ever Recorded”. I recommend acceptance pending a re-framing, i.e. that this is a 10-year review of the 2014-2017 event and that similar analyses of the 2024 event deserve similar comparison, at least of the temperature / DHW / satellite data (since gathering the monitoring data from so many different sources is likely what took 10 years!)

Response: While the third global coral bleaching event was the most severe and widespread damage to reefs when it concluded, we are sad to see that the ongoing fourth global coral bleaching event may turn out to be even worse. We concur with the reviewer and have changed the title and re-framed this paper in light of current events.

That said, we did not include any quantitative analyses of heat stress for the more recent event, because it requires fairly extensive additional treatment (for example, deciding when the event began, and when or if it has ended). Those analyses are the subject of a separate manuscript that has recently been submitted for consideration at *Nature* (Spady et al., attached as supplement). We hope this reassures the editor and reviewer that we indeed consider a comparison with the current and (we believe) ongoing global event to be important and worthwhile, but also that attempting to do so in the current manuscript would not do service either to our comprehensive analysis of the bleaching data from GCBE3, nor the heat stress data from GCBE4.

*“ While the third global coral bleaching event was unprecedented at the time, the observed impacts may be equaled or surpassed in subsequent events, including the current (and ongoing) fourth global coral bleaching event [67]. As ocean warming continues, coral loss and associated reef degradation around the world are nearly certain to accelerate.
” (lines 325-328)*

R2.3 I would also take the moment to appreciate the work that many NOAA scientists and technologists have done over many years to advance this work, and that I hope the NOAA datasets remain useful to scientists for years to come. In the current political climate, it's also worth noting that publishing this article quickly will be a record of the usefulness of NOAA's tracking and forecasting of coral bleaching events worldwide.

Response: We thank the reviewer for these comments, and all authors agree that it is important that NOAA and its scientist be adequately credited for this work. To that end, because the US Administration changed since this MS was submitted, the MS has had to undergo additional internal review at NOAA. In response to that review, we have made some minor changes in wording (e.g., “global warming” and “anthropogenic climate change” to “ocean warming”), and we reworded the final sentence of the main text, so that it could not be read as advocating for a particular policy response to the effects of climate change on coral reefs, while still emphasizing the likely consequences of ongoing warming for the status of the world's coral reefs.

Major suggestions:

R2.4 Title. Change to, “The 2014–17 Global Coral-Bleaching Event: A Record of Severe and Widespread Coral Reef Damage”

Response: We thank the reviewer for this suggestion, which we have re-worded as “Severe and widespread coral reef damage during the 2014-2017 Global Coral Bleaching Event”.

R2.5 Abstract. Identify Fourth Global Bleaching Event and how these findings can help inform continued predictions / understanding of bleaching events.

Response: Unfortunately, we were at 149 words and the abstract word limit is 150, so we could not incorporate this suggestion in the abstract. However, we do make changes in the main text as noted above. If the editor suggests we can exceed the 150-word limit to add language about the fourth global coral bleaching event, we would be glad to do so.

R2.6 Personally I find the ‘GCBE’ acronyms to be distracting and not a common acronym, I would suggest removing and replacing with ‘bleaching events’ or ‘Third Global Bleaching Event’, etc.

Response: We agree that excessive proliferation and use of acronyms and initialisms can be distracting. At the same time, we believe that GCBE is in common use, having been established in Little et al. (2022), and appearing in numerous other manuscripts of which we are aware. Moreover, they are useful shorthand in the figures we use comparing the different global coral bleaching events, and in sentences that refer to multiple global coral bleaching events (to avoid use of the same 4-word phrase 2 or 3 times in one sentence). Our

compromise has been to reduce use of GCBE in the main text, either writing out “third global coral bleaching event”, or, when clear from the context, “the 2015-2017 event” or similar, either in addition to or in place of the acronyms (e.g., lines 75, 86, 99, 107-116, 280, 314, 325-327).

R2.7 Fig1 - Any chance to add a panel of NOAA Alerts for the 2024 event to put these panels in context with the most recent event? I'm definitely not asking for any addition or extra analysis of the monitoring data / bleaching simulations - just simply for the thermal context of these events.

Response: As noted in our response to R2.2, we have not included this, due to the need for a more extensive and focused treatment of the fourth global event, which is found in the uploaded Spady et al manuscript.

Minor comments

R2.8 L106 - Was this 'consistent spatial evolution' also what you observed during this event? Or did this bleaching event follow a different spatial expansion? I suggest replacing 'evolution' with 'expansion' to be more appropriate (it's not an evolution over many generations, per se)

Response: We agree with the reviewer that the usage of the term 'evolution' is not consistent with the biological definition and are using the term “spatial progression” instead.